# SARS-CoV-2 S Glycoprotein Stabilization Strategies

**DOI:** 10.3390/v15020558

**Published:** 2023-02-17

**Authors:** Borys Pedenko, Guidenn Sulbaran, Delphine Guilligay, Gregory Effantin, Winfried Weissenhorn

**Affiliations:** University Grenoble Alpes, CEA, CNRS, Institut de Biologie Structurale (IBS), 38000 Grenoble, France

**Keywords:** SARS-CoV-2, S glycoprotein, stabilization, virus entry, neutralizing antibodies, vaccine

## Abstract

The SARS-CoV-2 pandemic has again shown that structural biology plays an important role in understanding biological mechanisms and exploiting structural data for therapeutic interventions. Notably, previous work on SARS-related glycoproteins has paved the way for the rapid structural determination of the SARS-CoV-2 S glycoprotein, which is the main target for neutralizing antibodies. Therefore, all vaccine approaches aimed to employ S as an immunogen to induce neutralizing antibodies. Like all enveloped virus glycoproteins, SARS-CoV-2 S native prefusion trimers are in a metastable conformation, which primes the glycoprotein for the entry process via membrane fusion. S-mediated entry is associated with major conformational changes in S, which can expose many off-target epitopes that deviate vaccination approaches from the major aim of inducing neutralizing antibodies, which mainly target the native prefusion trimer conformation. Here, we review the viral glycoprotein stabilization methods developed prior to SARS-CoV-2, and applied to SARS-CoV-2 S, in order to stabilize S in the prefusion conformation. The importance of structure-based approaches is highlighted by the benefits of employing stabilized S trimers versus non-stabilized S in vaccines with respect to their protective efficacy.

## 1. Introduction

Coronaviruses (CoVs) are positive-sense single-stranded RNA viruses composed of four genera (Alpha-, Beta-, Delta-, and Gammacoronaviruses), infecting mainly mammals and birds. Notably, the Betacoronaviruses, composed of four lineages, have emerged as important human pathogens. Although common cold viruses human coronavirus OC43 (HCoV-OC43) and human coronavirus HKU1 (HCoV-HKU1; lineage 1, Embecovirus) have been endemic for a long time, severe disease is associated with lineage 2 Sarbecoviruses (Severe Acute Respiratory Syndrome Coronavirus (SARS-CoV) and Severe Acute Respiratory Syndrome Coronavirus 2 (SARS-CoV-2)) [1]. Both viral outbreaks originated in China. While the spread of the SARS-CoV outbreak in 2003 was limited to four other countries, SARS-CoV-2 started to spread all over the world in 2019, causing an important pandemic with a heavy human toll (https://covid19.who.int/, accessed on 13 February 2023) and economic burden. The third highly pathogenic member, Middle Eastern Respiratory Syndrome coronavirus (MERS-CoV), is a member of lineage 3, Merbecovirus, whose transmission from camels to humans was far less successful, causing mostly local outbreaks in the Middle East. Bats are the natural reservoir hosts for the Betacoronaviruses SARS-CoV and SARS-CoV-2, and transmission to humans most likely occurred via intermediate hosts [2,3,4]. 

Much of the research on SARS-CoV-2 has focused on the development of vaccines since the beginning of the pandemic. To date, a number of different vaccines (Table 1) have been licensed by regulatory agencies. Notably, injections of more than thirteen billion vaccine doses worldwide (https://covid19.who.int/, accessed on 13 February 2023) have substantially lowered the risk for severe disease. However, the rapid emergence of highly mutated viruses (variants of concern, VOC) [5] has severely diminished vaccine protection, leading to many breakthrough infections. 

An important correlate of protection is the generation of neutralizing antibodies upon vaccination or infection [6,7,8]. The viral glycoprotein S is the main target for neutralizing antibodies, and thus is the main component of current vaccines. Here, we review the contribution of the structural biology of S to understanding its role in the virus’s life cycle, as well as its application in approved and ongoing vaccine strategies in preclinical settings. We further discuss the S structure in light of the rapidly emerging variants that, together with S structures in complex with broadly neutralizing antibodies (bnAb), explain immune evasion.

## 2. Structural Biology of the S Glycoprotein

Structural biology on the SARS-CoV-2 S glycoprotein was facilitated by previous structural studies of S glycoproteins from HCoV-HKU1 [23], SARS-CoV, MERS-CoV [24,25], mouse coronavirus [26], human coronavirus NL63 (HCoV-NL63) [27], porcine deltacoronavirus (PDCoV) [28,29,30], and human coronavirus HCoV-229E [31], and revealed the overall organization of the S trimers. A strategy to stabilize S via proline mutations, which was developed within these structural studies, demonstrated increased resistance to conformational changes induced by receptor recognition [32]. Consequently, this established S design allowed the ectodomain structures of SARS-CoV-2 S to be rapidly determined at the onset of the pandemic [33,34] (Figure 1). S is a type I membrane protein that is highly glycosylated and organized into two subunits: the receptor-binding subunit (S1) and the fusion subunit (S2). S1 contains an N-terminal domain (NTD); a receptor-binding domain (RBD); and two C-terminal domains (CTD1 and CTD2), also denoted as subdomains 1 and 2 (SD1 and SD2). S2 harbors the fusion peptide (FP), the fusion peptide proximal region (FPPR), heptad repeat 1 (HR1), the central helix (CH), the connector domain (CD), the heptad repeat 2 (HR2), the transmembrane domain (TM), and a short cytoplasmic tail (CT) (Figure 1A). Subsequently, structures of isolated full-length SARS-CoV-2 S in the prefusion and post-fusion conformations [35,36], and of S anchored in the virus envelope, were reported [37,38,39,40]. The S protomer adopts an overall Y-shaped form in its prefusion conformation, with the two arms formed by NTD and RBD-CTD1, and the extension by CTD2 and S2 (Figure 1B,D). S1 wraps around the central helix of S2, positioning the S2 HR1 towards the viral membrane. Significant conformational variability is only observed for the positioning of the RBD domain at the trimer apex. The RBD interacts with the major cellular receptor, angiotensin-converting enzyme 2 (ACE2), on host cells [41,42]. In addition to ACE2, a number of auxiliary receptors and/or cofactors have been reported to play a role in entry, dependent on the tissue (reviewed in [43]). All three RBD domains are either in the down position (pdb 6XR8) (Figure 1B,D,F), or one, two, or all three are in the open-up position (one 7KRR RBD in the up position; two 7EB5 RBDs in the up position; and three 7KML RBDs in the up position) (Figure 1F), allowing interaction with one, two, or three ACE2 receptors [44,45,46,47]. The NTD is located at the periphery of the trimer, contacting the adjacent RBD, while CTD1 and CTD2 interact with S2. 

SARS-CoV-2 enters cells via endocytosis and fuses the viral membrane with late endosomal membranes. Fusion is triggered both by the low pH of the endosomes and additional proteolytic cleavage by the serine protease TMPRSS2 [42]. SARS-CoV-2 S is a typical class I membrane fusion protein, whereas S2 undergoes large conformational changes from the prefusion [35] to the post-fusion conformation, harboring a six-helical bundle structure [35,51] (Figure 1C,E). The class I architecture has been predicted by earlier coronavirus core post-fusion structures [52,53] via their similarity with other class I fusion proteins [54]. 

## 3. S and Antibody Recognition

Antibody-mediated neutralization targeting S, by either blocking receptor binding or blocking conformational changes, is the central defense strategy, which can include targeting the fusion machinery or opening-up/inactivating the S trimer [55]. Major antigenic sites are located in S1, the RBD, the NTD, and CTD1 (also named SD1, subdomain 1) [56,57,58,59,60,61,62]. The RBD harbors four major epitope classes that are recognized by a diverse set of neutralizing antibodies [63,64,65]. Class I and class II neutralizing antibodies recognize the ACE2 binding region, blocking ACE2 interaction [63]. Class I neutralizing antibodies bind RBDs in the “up” conformation and only block ACE2 binding, while class II neutralizing antibodies block ACE2 binding with RBDs either in the “up” or “down” conformation. Class III neutralizing antibodies block ACE2 binding in the “up” and “down” RBD conformations and they can, in addition, interact with adjacent RBD protomers. Class IV neutralizing antibodies do not interfere with ACE2 binding, but recognize conserved epitopes in the RBD “up” conformation. Furthermore, some class IV epitope-neutralizing antibodies have broad neutralizing activity against different SARS-CoV-2 variants [66,67], including Omicron [68,69,70], as well as related coronaviruses [63,64,65,71]. Major neutralizing epitopes, are also present in the NTD [60,72,73,74,75,76]. NTD was suggested to bind lectin receptors, which in turn may act as alternative entry receptors [67,77]. S2 is also immunogenic and harbors neutralizing epitopes [78]. Notably, S2 stem helix-recognizing antibodies that have broad neutralizing activity against all SARS-related viruses, as well as human Betacoronaviruses, have been isolated [79,80,81,82].

## 4. Viral Glycoprotein Stabilization Strategies

Class I viral glycoproteins are composed of at least two domains: a receptor-binding domain and a fusion protein domain that anchors the glycoprotein to the cellular membrane. They are generally expressed as precursor proteins that are cleaved by cellular proteases (such as furin) into the two subunits, thereby placing the hydrophobic fusion peptide at or close to the N-terminus of the fusion protein subunit. Furin or protease cleavage renders the glycoprotein metastable and activates its fusion potential, which allows it to exert two main functions. First, the receptor-binding domain of the glycoprotein attaches the virion to the target cell membrane. Second, receptor binding induces either the direct fusion of the virus membrane with the plasma membrane of target cells, as in the case of HIV-1, or permits virus entry via the endosomal pathway, where the low pH of the endosome triggers the membrane fusion activity of the glycoprotein. The latter is the case for SARS-CoV-2 S-mediated entry, which requires a second proteolytic cleavage in the endosome triggered by S1 shedding and S2 cleavage triggered by cathepsin L [83,84]. Receptor binding, followed by a low-pH environment and a second proteolytic cleavage in the case of SARS-CoV-2 S, triggers extensive conformational changes in the glycoprotein, especially in the fusion protein subunit. The latter folds into a stable post-fusion conformation positioning the transmembrane region and the fusion peptide at the same end of a rod-like structure [35,51]. The role of the conformational change is to bring the viral and cellular membranes into close proximity in order to catalyze membrane fusion [54]. 

Viral glycoproteins are metastable and need to be stabilized in order to keep them in the prefusion conformation, which can switch spontaneously into the post-fusion state. This may be triggered by high temperatures and ionic conditions, or may be favored by a delicate equilibrium between “closed” and “open” states that eventually facilitates complete fusion protein rearrangement, as described for SARS-CoV-2 S2 [85]. The cold sensitivity of S was further identified as another source of conformational lability [86]. Since vaccines aim to induce neutralizing antibodies that recognize the prefusion conformation of the viral glycoprotein, several strategies can be employed to stabilize the viral glycoprotein in the prefusion conformation in order to prevent the switch to the post-fusion state or any other abnormal non-functional conformation, which may induce mostly non-neutralizing antibodies with little or no protective effect. 

Stabilizing mutations were first introduced into the HIV-1 envelope glycoprotein (Env) with the aim to preserve the native Env trimer conformation. This started with engineering a disulfide bond between gp120 and gp41, to prevent gp120 shedding from cleaved Env trimers; however, this alone was not sufficient for preventing trimer disassembly [87]. The additional introduction of a proline at the Ile559 position (I559P) then led to the first stabilized prefusion Env trimer, named SOSIP gp140 [88]. The solubility of this stabilized Env version was then further improved by removing the hydrophobic gp41 MPER, resulting in BG505 SOSIP.664 [89]. BG505 SOSIP.664 has been extensively structurally characterized [90,91,92], and its immunogenicity was evaluated in preclinical studies, which demonstrated autologous virus neutralization upon immunization [93]. Since then, a plethora of stabilized Envs from different HIV-1 clades have been reported. They are all based on the SOSIP.664 design and include additional disulfides, cavity filling mutations, and other proline mutations, all leading to enhanced thermostability of the Env trimers [94,95,96,97,98,99,100,101,102]. 

An alternative approach to stabilizing the HIV-1 Env prefusion BG505 SOSIP.664 trimer conformation is based on chemical cross-linking. Cross-linking HIV-1 Env BG505 SOSIP with glutaraldehyde [103] introduced two specific cross-links, one between gp120 monomers at the trimer apex and another between gp120 and gp41 at the trimer interface, thereby enhancing thermostability and immunogenicity [104].

## 5. S Stabilization Strategies

SARS-CoV-2 S was first stabilized by two proline mutations within S2 (K986P and V987P), named S ‘2P’, which enhanced stability and allowed structural solution [33] (Figure 2A). The stability of S ‘2P’ is largely limited over time [33], but was further improved by adding four more proline substitutions in the S version named ‘6P’ (K986P, V987P, F817P, A892P, A899P, and A942P), which increased the thermostability of S to 50 °C [105] (pdb 6XKL) (Figure 2A). The positive stabilizing effect of residue changes A892P and A942P was further confirmed by another study that introduced the same changes in A892P and A942P in combination with D614N, R682S, R685G, and V987P mutations [106] (pdb 7A4N) (Figure 2A). 

A second approach to stabilizing S employed chemical cross-linking of SARS-CoV-2 S ‘2P’ by formaldehyde treatment, which introduced specific cross-links between RBD protomers at residues K378 and R408 and between S2 subunit residues K947 and K776 and/or R1019 of two adjacent protomers (Figure 2B). Introducing these covalent linkages between protomers prevented trimer dissociation and locked the native S trimer in the closed RBD-down conformation, thereby increasing its thermostability to 65 °C without affecting overall immunogenicity [107].

A third approach to stabilizing S was via engineering new disulfide bonds within SARS-CoV-2 S ‘2P’, at positions Ser 383 and Asp 985 and Gly 413 and Val 987 [108] (Figure 2C). Moreover, linoleic acid binding into a pocket within the RBD (Figure 2D) renders S more stable [109,110]. Employing deep mutational scanning (DMS) data identified mutations to fill the linoleic acid-binding pocket in the RBD, which in turn increased the expression yield, as well as the thermal stability of S [111]. 

In general, proline substitutions have been proposed in order to interfere with helix formation within predicted coiled coil regions of viral fusion proteins, thereby blocking the conformational changes required for folding into the post-fusion conformation. Notably, the latter transition is absolutely required for virus entry via the fusion of viral and cellular membranes. Therefore, the introduction of proline substitutions has since evolved as a general stabilization technique for viral class I glycoproteins. Stabilizing proline substitutions have since been introduced into the F glycoprotein from Paramyxoviruses, such as the respiratory syncytial virus (RSV) [112] and human metapneumovirus (hMPV) [113], the Lassa virus (LASV) glycoprotein complex (GPC) that was additionally stabilized by a G1-G2 interdomain disulfide bond [114], Ebola and Marburg virus glycoproteins [115], and the influenza virus hemagglutinin glycoprotein [116]. 

## 6. Importance of Glycoprotein Stabilization for Vaccine Development

It has long been recognized that the conformation of the immunogen used for vaccination is important for the induction of neutralizing antibody responses. A prime case is, again, the HIV-1 Env glycoprotein, where the majority of antibodies produced during natural infection are directed to non-neutralizing epitopes [117,118] that are associated with the recognition of the off-target conformational states of the glycoprotein [119]. It is therefore important to assess the effect of glycoprotein stabilization that is the preservation of the native prefusion conformation, in light of the protective efficacy of currently licensed SARS-CoV-2 vaccines. A direct comparison of vaccine efficacy is complicated by the different technical approaches used to measure overall antibody responses and neutralizing antibody responses to these vaccines, which vary substantially [120,121]. However, a number of preclinical studies showed the benefit of the S ‘2P’ mutations over the unmodified wild-type sequence, with regard to immunogenicity and protection from infection via neutralizing antibody titers. This was first confirmed for the MERS-CoV S glycoprotein [25], SARS-CoV-2 S glycoprotein vaccines [122], AD26-based vectors expressing S variants [18,123], and the Sanofi mRNA vaccine MRT5500 [124]. 

Notably, a comparison of the overall antibody reactivity against prefusion versus post-fusion conformations, and the induced neutralizing antibody activity of clinically approved vaccines carrying the S ‘2P’ stabilization versus S vaccines without stabilization (Table 1), confirmed the positive effect of S ‘2P’ stabilization. These analyses were performed with human sera from individuals that were not exposed to SARS-CoV-2 before being immunized with two doses of the Moderna/NIAID mRNA-1273, Pfizer/BioNTech Comirnaty, Novavax NYX-CoV2373, Janssen /Johnson & Johnson Ad26.CoV2.S or Astra-Zeneca/Oxford AZD1222, Gamaleya Sputnik V and Sinopharm CoVilo/BBIBP-CorV vaccines (Table 1). Superior immune responses with the S ‘2P’-stabilized vaccines over the non-stabilized native S vaccines were reported, based on the correlation between antibody binding titers and neutralization potency, as well as the contribution of RBD- and NTD-specific antibodies to the cross-neutralization of SARS-CoV-2 variants [125]. Neutralizing antibody titers were also positively correlated with NTD- and RBD-specific binding titers, which is in line with both domains acting as the main target for the generation of neutralizing antibodies upon infection and vaccination [107,126,127,128].

However, it should be noted that non-stabilized S glycoprotein vaccines have been shown to provide protection as well, albeit at lower efficacy. This is in line with the abovementioned immune dominance of the RBD, which is most likely sufficiently exposed in non-native conformations of S. Indeed, numerous vaccine candidates composed of only RBDs demonstrated the significant immunogenicity and generation of neutralizing antibodies [129,130,131,132,133,134,135,136,137,138,139,140]. Furthermore, since the RBD contains conserved epitopes that are recognized across most circulating clades, specifically targeting conserved epitopes within the RBD may generate SARS-CoV-2 vaccines with broader cross-variant neutralization and thus protective activity [141,142]. This is in line with the findings that the cross-neutralization of SARS-CoV-2 variants is determined by RBD-specific antibodies [125] and that RBD-targeting neutralizing antibody cocktails have the potential to protect their recipients from infection with variants of Omicron [143].

## 7. S of SARS-CoV-2 Variants

Since the outbreak in December 2019, the original Wuhan strain has mutated and generated a series of new variants, named Alpha, Beta, Epsilon, Eta, Iota, Kappa, Delta, Lambda, Gamma, Zeta, Theta, and Omicron, which spread all over the world. As of the end of 2022, the current prevailing strains are Omicron variants [5,144,145,146]. The pandemic has thus reminded us of the enormous capacity of a virus to adapt to the selective pressure imposed by the immune system. Notably, as of November 2022, more than 13 billion COVID-19 vaccine doses have been administered worldwide (https://covid19.who.int/, accessed on 13 February 2023). The longitudinally acquired mutations have been mostly associated with higher transmissibility, without introducing major structural changes within S [85,147,148]. Omicron is also, to date, the most mutated version, with 32 mutations in the S glycoprotein, among which 14 are present in all Omicron variants [149,150]. Consequently, many of the amino acid changes led to the modulation of the antibody epitopes, which in turn are no longer recognized by most NTD- and RBD-specific neutralizing antibodies, thereby facilitating the viral immune escape associated with Omicron infection [151,152]. However, S did not become completely resistant to the recognition by neutralizing antibodies, since potent and broad RBD-specific antibodies [61], as well as NTD-specific antibodies targeting supersites [73] that overcome resistance, have been isolated. Although first-generation vaccines show limited protection against Omicron variants [153,154,155,156], the immune response can be boosted with mRNA vaccines [157,158], and vaccination was shown to prevent severe disease. Notably, Omicron variants cause milder disease in small animal models, which suggests that the mutations not only led to antibody escape, but also attenuated these SARS-CoV-2 variants [159].

## 8. Conclusions

Significant progress has been made in understanding the structure of the S glycoprotein and the conformational changes catalyzing virus entry via membrane fusion. The metastability of S has been addressed by different approaches, notably using proline mutations in the S2 subunit that stabilize S in the native prefusion conformation. The positive effect of stabilization has since been proven by demonstrating superior neutralizing antibody responses upon vaccination with stabilized S glycoproteins versus non-stabilized native S. A major challenge of the field is now to develop vaccine approaches that will provide broad and potent protection against current and newly arising SARS-CoV-2 variants. This will likely be feasible, since a number of highly conserved epitopes in S have been identified, and future vaccine approaches can focus the immune response on these conserved epitopes. Another major challenge is based on developing vaccines that will provide mucosal protection to prevent the virus from spreading [160,161,162].

## Figures and Tables

**Figure 1 viruses-15-00558-f001:**
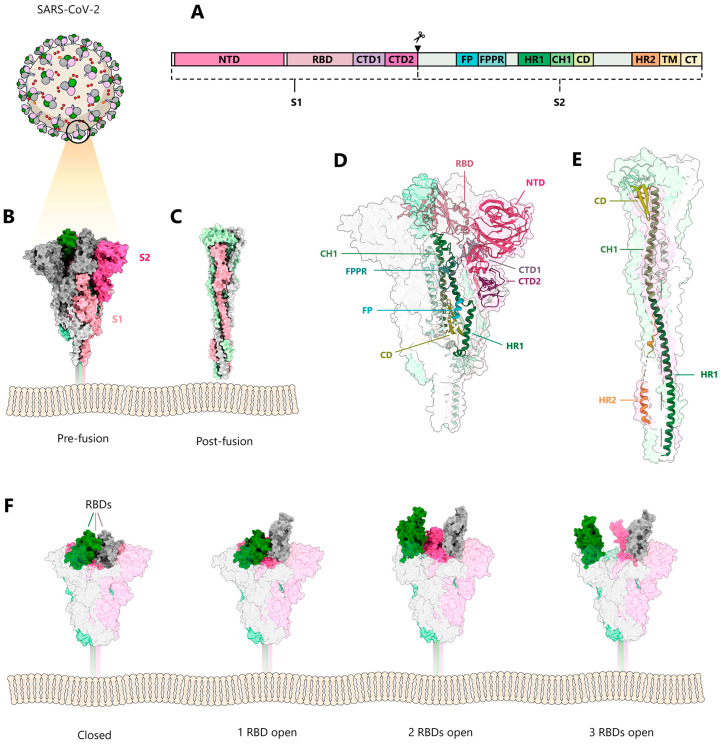
Conformations of the SARS-CoV-2 spike protein. (**A**) Model of SARS-CoV-2 and schematic representation of the domain structure of the spike glycoprotein S. (**B**) Side-view of the prefusion S trimer (PDB:6XR8) [35], showing the three protomers in grey, pink, and green, with the corresponding S1 subunits colored lighter than the S2 subunits. (**C**) Side-view of the post-fusion trimer (PDB:6XRA) [35]. (**D**) Side-view of the prefusion and (**E**) the post-fusion trimers shown as molecular envelope, with a single protomer illustrated as ribbon; the domains are colored as indicated in (**A**) (PDB:6XRA and 6XR8) [35]. (**F**) Conformational variability of RBDs in the prefusion spike; from left to right: closed (PDB: 6XRA) [35], one receptor-binding domain (RBD) in the up position (PDB: 7KRR) [48], two RBDs in the up position (PDB: 7EB5) [49], and three RBDs in the up position (PDB: 7KML) [50].

**Figure 2 viruses-15-00558-f002:**
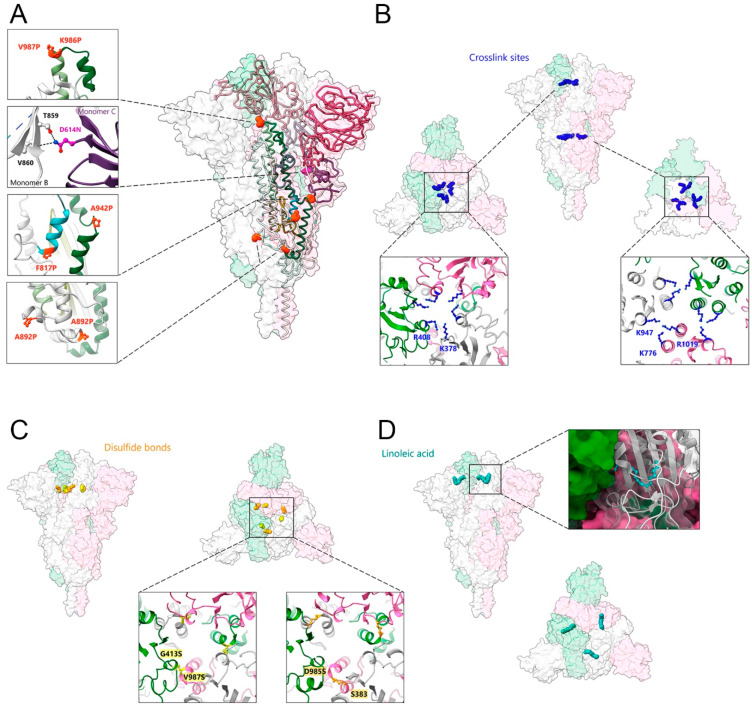
Strategies of SARS-CoV-2 spike protein stabilization. (**A**) Introduction of stabilizing mutations. Close-ups of the orange spheres represent proline substitutions K986P, V987P, F817P, A892P, A899P, and A942P (PDB:6XKL) [105]. Close-up of the violet spheres corresponding to the mutation of D614 to asparagine, which directly contacts T859 and V860 from the adjacent protomer; residue mutations at positions R82S and R685G are disordered (PDB:7A4N) [106]. (**B**) Chemical crosslinking by formaldehyde mediates the formation of covalent bonds between adjacent protomers by residues K378 and R408 (close-up to the **left**) and by residues K947 and K776 and/or R1019 (close-up to the **right**) (PDB: 7Q1Z) [107]. (**C**) The introduction of disulfide bonds between adjacent protomers at positions G413C and V987C (close-up to the **left**; PDB: 6ZOX) and at S383C and D985C (close-up to the **right**) (PDB: 6ZOY) [108]. (**D**) The binding of linoleic acid (in cyan blue) into an RBD pocket stabilizes the spike protein in the prefusion conformation (close-up to the right shows one of the RBDs in the ribbon accommodating linoleic acid) (PDB: 6ZB5) [109].

**Table 1 viruses-15-00558-t001:** SARS-CoV-2 vaccines authorized by national regulatory agencies.

S ‘2P’ Stabilization	Ref.	S Wild-Type	Ref.
Moderna/NIAID, mRNA-1273	[9,10]	Astra-Zeneca/Oxford, AZD1222	[11]
Pfizer/BioNTech, Comirnaty (BNT162b2)	[12,13]	Gamaleya, Sputnik V	[14]
Novavax NYX-CoV2373	[15,16]	Sinopharm, CoVilo/BBIBP-CorV	[17]
Janssen/Johnson & Johnson, Ad26.CoV2.S	[18]	CanSino Biologics, Convidencia	[19]
Sanofi/GSK, VidPrevtyn^®^ Beta	[20]	Sinovac, CoronaVac	[21]
		Bharat Biotech/Indian Council of Medical Research, Covaxin	[22]

## Data Availability

Not applicable.

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
