# Peer review of "SARS-CoV-2 S Glycoprotein Stabilization Strategies"

_viruses, 2023, doi:10.3390/v15020558_

Round 1

Reviewer 1 Report

The manuscript summerized all the  stabilization methods used to stabilize SARS-CoV-2 spike protein in the prefusion conformation. The aim is very interesting but there are some typing errors and some confusing sentences. Therefore, the authors should revise carefully all the text.

1. Introduction is missing;

2. All viral names must be written first with the extended form and then with their acronym. In addition, HCoV-OC43, HCoV-HKU1, SARS-CoV, SARS-CoV-2, have to be written properly;

3. I don't understand what are bnAbs and S "2P", since they are not explained in the text.

Author Response

  1. Introduction is missing;

This has been reformatted.

  1. All viral names must be written first with the extended form and then with their acronym. In addition, HCoV-OC43, HCoV-HKU1, SARS-CoV, SARS-CoV-2, have to be written properly;

This has been corrected in the introduction.

  1. I don't understand what are bnAbs and S "2P", since they are not explained in the text.

We have defined the abbreviations:

Page 2:

We further discuss the S structure in light of the rapidly emerging variants that together with S structures in complex with broadly neutralizing antibodies (bnAb) explain immune evasion.

Page 5:

  1. S stabilization strategies

SARS-CoV-2 S was first stabilized by two proline mutations within S2 (K986P and V987P), named S ‘2P’, that enhanced stability and allowed structure solution

Reviewer 2 Report

The review manuscript entitled “SARS CoV-2 S glycoprotein stabilization strategies” presented valuable, attractive, and comprehensive information about the biology, pathology, and physiology of the SARS-CoV-2 virus. Importantly, the authors provided a hot topic idea that could be attractive to many researchers. Also, the manuscript was designed properly and had a very good English style. After carefully reading the manuscript, I believe that it can be accepted in its present form. There is just one comment that the authors should consider in the revised version.

-Angiotensin-converting enzyme 2 had introduced two times. Please check it out.

Author Response

-Angiotensin-converting enzyme 2 had introduced two times. Please check it out.

This has been corrected.

Round 2

Reviewer 1 Report

The authors properly improved the quality of the review, that is now suitable for publication.